# Punishment insensitivity emerges from impaired contingency detection, not aversion insensitivity or reward dominance

Philip Jean-Richard-dit-Bressel[1], Cassandra Ma[1], Laura A Bradfield[1,2,3], Simon Killcross[1], Gavan P McNally[1]*

[1]School of Psychology, UNSW Sydney, Sydney, Australia; [2]Centre for Neuroscience and Regenerative Medicine, Faculty of Science, University of Technology Sydney, Sydney, Australia; [3]St Vincent's Centre for Applied Medical Research, Sydney, Australia

**Abstract** Our behaviour is shaped by its consequences – we seek rewards and avoid harm. It has been reported that individuals vary markedly in their avoidance of detrimental consequences, that is in their sensitivity to punishment. The underpinnings of this variability are poorly understood; they may be driven by differences in aversion sensitivity, motivation for reward, and/or instrumental control. We examined these hypotheses by applying several analysis strategies to the behaviour of rats ($n$ = 48; 18 female) trained in a conditioned punishment task that permitted concurrent assessment of punishment, reward-seeking, and Pavlovian fear. We show that punishment insensitivity is a unique phenotype, unrelated to differences in reward-seeking and Pavlovian fear, and due to a failure of instrumental control. Subjects insensitive to punishment are afraid of aversive events, they are simply unable to change their behaviour to avoid them.

*For correspondence:
g.mcnally@unsw.edu.au

**Competing interests:** The authors declare that no competing interests exist.

## Introduction

Our behaviours, decisions, and choices are shaped by their consequences. When rewarded, they are likely to be repeated, but when punished they are not. Reward and punishment are among the most fundamental psychological building blocks of behaviour. They allow us to cope with a changing world, maximising our probability of survival by seeking utility and avoiding harm. Yet there is often pronounced variation between individuals in responsivity to the consequences of their behaviours. Notably, individuals differ significantly in their sensitivity to punishment (*Corr, 2004*; *Corr, 2013*; *Gray, 1970*; *Gray, 1982*; *Marchant et al., 2018*). Insensitivity to punishment is observed experimentally as impaired suppression of behaviours that cause aversive events. Punishment sensitivity plays an important role in normal learning, decision making as well as emotion (*Corr, 2004*). Differences in sensitivity to punishment have been implicated in the aetiology or maintenance of a range of psychopathologies including conduct disorder (*Briggs-Gowan et al., 2014*; *Dadds and Salmon, 2003*), drug and behavioural addictions (*Vanderschuren et al., 2017*), eating disorders (*Monteleone et al., 2018*), psychopathy (*Blair et al., 2006*; *Gregory et al., 2015*), and depression (*Elliott et al., 1996*; *Eshel and Roiser, 2010*). Moreover, punishment sensitivity is an increasingly popular measure of the motivation to engage in drug-seeking and drug-taking (*Augier et al., 2018*; *Deroche-Gamonet et al., 2004*; *Kasanetz et al., 2013*; *Marchant et al., 2018*; *Pascoli et al., 2015*; *Vanderschuren and Everitt, 2004*; *Vanderschuren et al., 2017*).

The cause(s) of differences in punishment sensitivity are poorly understood. Three main mechanisms have been proposed (*Figure 1*). First, individual differences in punishment learning may be

**Figure 1.** Potential sources of punishment insensitivity.

due to temperamental differences in aversive valuation or aversion sensitivity (*Corr, 2004*; *Gray, 1970*). Successful punishment learning requires that the punisher be encoded as aversive. If individuals differ in the extent to which they are sensitive to the aversiveness of punishers, then they are likely to differ in the extent to which they will suppress any behaviour that produces a punisher. A second possibility is that punishment insensitive individuals show reward dominance, with choices and behaviour more strongly determined via the value of any rewards they earn rather than any punishment they incur (*O'Brien and Frick, 1996*; *Robinson and Berridge, 2003*). A final, and not mutually exclusive, possibility is that individual differences in punishment sensitivity emerge from individual differences in aversive instrumental learning and control (*Seligman, 1970*). Punishment learning involves encoding the instrumental contingency between behaviour and its adverse consequences. It is possible that punishment insensitive and sensitive individuals may encode the punisher as equally aversive but differ in their ability to detect or encode the contingency between their behaviour and the punisher and/or in their ability to control behaviour according to this instrumental knowledge.

Mechanistic behavioural assessment of differences in punishment sensitivity is difficult. One way of distinguishing between these different mechanisms is to examine responses to the punisher directly. However, the magnitude of the unconditioned response to a stimulus has little bearing on what is learned about that stimulus (*Rescorla, 1988*). An alternative approach is to use tasks that dissociate reward and aversion learning to reveal what relationships exists between them, thus allowing diagnosis of common origins. For example, if individual differences in punishment sensitivity are due to differences in aversive valuation or sensitivity (*Corr, 2004*; *Gray, 1970*), then this should be reflected in other forms of learning about the same aversive event in the same individuals, such as Pavlovian conditioning. So, insight into the origins of differences in punishment sensitivity could be obtained through comparisons of instrumental reward learning, instrumental punishment learning, and Pavlovian fear learning. However, there are methodological issues involved when making such assessments. For example, in order to understand individual variation, these different forms of learning have to be assessed in the same individuals. To avoid carry-over effects, they must be assessed concurrently. Finally, the same measure should be used to quantify each form of learning. Few tasks solve each of these methodological issues and none have been used to understand individual differences in punishment learning.

Here we used a conditioned punishment task that permitted us to concurrently identify and study individual differences in instrumental reward learning, instrumental punishment learning, as well as Pavlovian fear learning using the same behavioural measure (*Killcross et al., 1997*). Rats were trained to respond on two levers for food reward. We then introduced concurrent punishment and Pavlovian fear contingencies on one lever but not the other. We used rates of lever pressing as our measure of reward, punishment, and fear. In addition to direct comparisons of lever pressing performance between the three contingencies, we used a variety of data analytic strategies (multidimensional scaling, principal components analysis, factor analysis, and k-means clustering) to understand the relationship between individual differences in instrumental reward learning, instrumental punishment learning, and Pavlovian fear learning. If punishment insensitivity is attributable to differences in

reward dominance, then individual differences in punishment learning should be related to differences in instrumental reward seeking. If punishment sensitivity is related to aversive insensitivity, then individual differences in punishment and fear should be related to each other. Finally, if individual differences in punishment insensitivity are attributable to punishment-specific deficits, then no relationship between the three forms of learning should be apparent.

## Results

### Individual differences in punishment and fear

Three contingencies were in effect within this task: the instrumental contingency of reward which should maintain responding on both levers; the instrumental contingency of punishment, which should bias animals away from the punished response (i.e. punishment suppression), and the aversive Pavlovian contingency that drives fear conditioning to predictive cues and suppresses ongoing behaviour (i.e. Pavlovian suppression). Each of these three effects were observed (*Figure 2*). Prior to aversive training, there was no preference between pressing the to-be-punished versus unpunished lever ($F(1,47) = .071$; $p = 0.791$, $\eta_p^2$ 0.001) (*Figure 2*). Across the course of aversive training, reward learning was maintained and punishment learning as well as fear learning were observed. There was

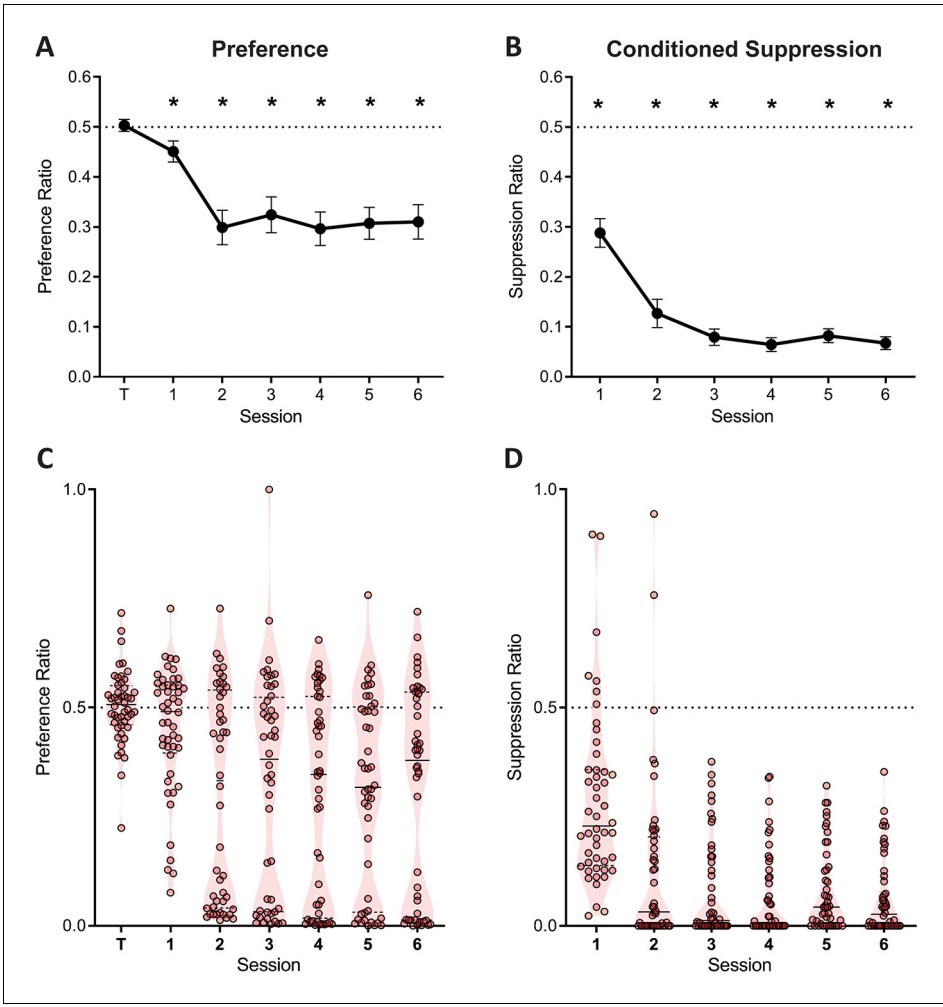

**Figure 2.** Lever preference and conditioned suppression across conditioned punishment. (A) Mean ± SEM preference ratios showing evidence for punishment. (B) Mean ± SEM suppression ratios showing evidence for fear. (C) Violin plots and individual subject preference ratios. (D) Violin plots and individual subject conditioned suppression ratios.

evidence for punishment learning because there was less lever pressing on the punished lever than the unpunished lever (the traditional measure of punishment learning in this task) (*Figure 2A*). Punishment avoidance increased across days (linear trend: F (1,47)=7.49; p=0.009, $\eta_p^2$ 0.137). Follow-up analyses revealed significant punishment suppression for each session (1st session: F (1,47)=5.48; p=0.024, $\eta_p^2$ 0.137; remaining sessions: F (1,47)>23.4; p<0.001, $\eta_p^2$ 0.332). There was also robust evidence for Pavlovian fear (*Figure 2B*). Conditioned suppression elicited by presentations of the CS + also increased across training (F (1,47)=35.1; p<0.001, $\eta_p^2$ 0.427), with significant suppression being observed for each session (all F (1,47)>54.7; p<0.001, $\eta_p^2$ 0.537). However, as expected, these group-averaged data obscured pronounced individual differences. *Figure 2C and D* show the same data plotted at subject level. Punishment suppression appeared to be bimodally distributed with some subjects showing strong punishment suppression (i.e. punishment sensitivity) and others weaker or no punishment suppression (i.e. punishment insensitivity) (*Figure 2C*). There was also, albeit less pronounced, variation in Pavlovian fear (*Figure 2D*).

The evidence for punishment is derived from a preference ratio (responses on the punished lever relative to total responses on both levers). This measure is simple and valid, but it obscures the degree to which preferences are driven by changes in punished responding, unpunished responding, or both. Moreover, subjects that suppress punished and unpunished lever-pressing equally would have ratios of 0.5, which might mistakenly be interpreted as an absence of punishment avoidance and hence punishment insensitivity. Therefore, we also assessed suppression of punished and unpunished responding separately against pre-punished rates of responding (*Figure 3A*). Here a suppression ratio of 0.5 indicates no difference in rate of pressing relative to last day of training (i.e. punishment insensitivity) whereas a ratio of 0 indicates complete suppression. At the group level, this analysis showed a main effect of lever (F(1,47) = 44.39; p<0.001, $\eta_p^2$ 0.485), session (linear: F (1,47) = 9.476; p=0.003, $\eta_p^2$ 0.167), and a significant lever x session interaction (F(1,47) = 10.62; p=0.002, $\eta_p^2$ 0.184). This interaction was driven by a significant increase in the unpunished suppression ratio across sessions (linear: F(1,47) = 24.54; p<0.001, $\eta_p^2$ 0.343), but no significant change in punished lever suppression (linear: F(1,47) = .210; p = 0.649, $\eta_p^2$ 0.004). Punished lever suppression was significantly greater than unpunished lever suppression for all sessions (F(1,47) > 9.22; p<0.004, $\eta_p^2$ 0.164). So, robust punishment was observed using this measure. However, this group level analysis again obscured pronounced individual differences (*Figure 3B*). Examination of individual subject performances showed that suppression of responding on the punished lever, but not the unpunished lever, appeared bimodal (*Figure 3B and C*).

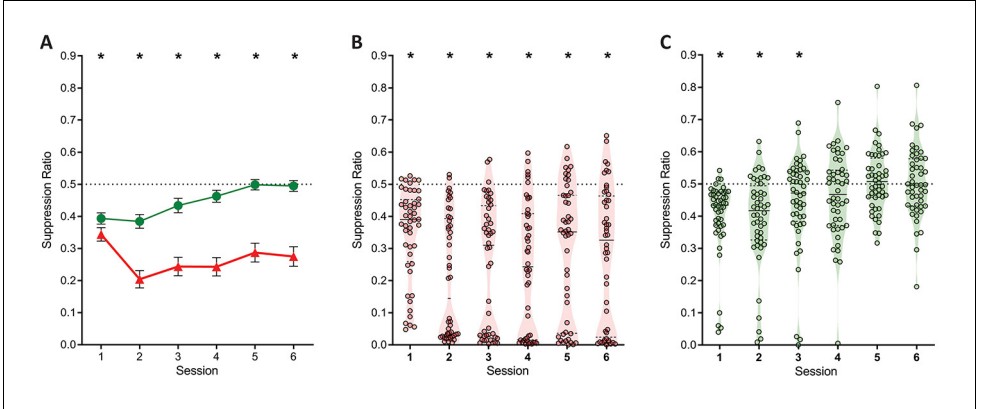

**Figure 3.** Lever-press suppression across conditioned punishment. (A) Mean ± SEM suppression ratios for responding on the punished (red) and unpunished (green) levers relative to training. *p<0.05 punished vs. unpunished. (B) Violin plots and individual subject suppression ratios for the punished lever. *p<0.05 punished vs. null ratio (0.5). (D) Violin plots and individual subject suppression ratios for the unpunished lever. *p<0.05 punished vs. null ratio (0.5).

## Punishment, reward, and fear are separate

To further examine the relationship between punishment, fear, and reward we examined the correlations between suppression on the punished lever (punishment), unpunished lever (reward), and conditioned suppression elicited by the CS+ (fear) across training (*Figure 4A*). We observed strong positive correlations among each of these measures across sessions, showing that each subject's relative behaviour was stable across days. However, we observed few significant positive correlations between the measures. Notably, for fear and punishment, the only significant correlation was negative and present only on the first day of training.

The correlation matrix is useful in visualising and understanding the relationship between measures but does not readily reveal meaningful, underlying dimensions that may explain overall similarities and dissimilarities. To visualise these, we used multidimensional scaling (*Figure 4B*). This showed that punishment, fear, and reward clustered in separate spaces. Punished lever suppression was clustered in a separate space from conditioned suppression, suggesting that punishment and fear are highly dissimilar to each other. Unpunished lever suppression was initially closely related to punished response suppression but became progressively different as training progressed.

These results show qualitative differences between punishment, reward, and fear. To better understand the shared variance between our measures we used Principal Components Analysis (PCA) to identify any shared underlying components in learning (*Figure 5A*). If there is a common aversion sensitivity that underpins punishment and fear learning, or any other common process, then PCA should identify it as a component with strong loadings from both punishment suppression and CS+ suppression. A 4-component solution was optimal, accounting for 75.9% of overall variance (*Figure 5—figure supplement 1*), with most measures well captured by these four components (*Figure 5A*, *Figure 5—figure supplement 1*). The first component captured the influence of punishment: punishment suppression across the course of training loaded strongly on this component. The second component captured the influence of contextual fear learning early during training: both initial punishment suppression and unpunished responding loaded positively on this component whereas CS+ suppression loaded negatively. The third component captured specific CS+ fear from

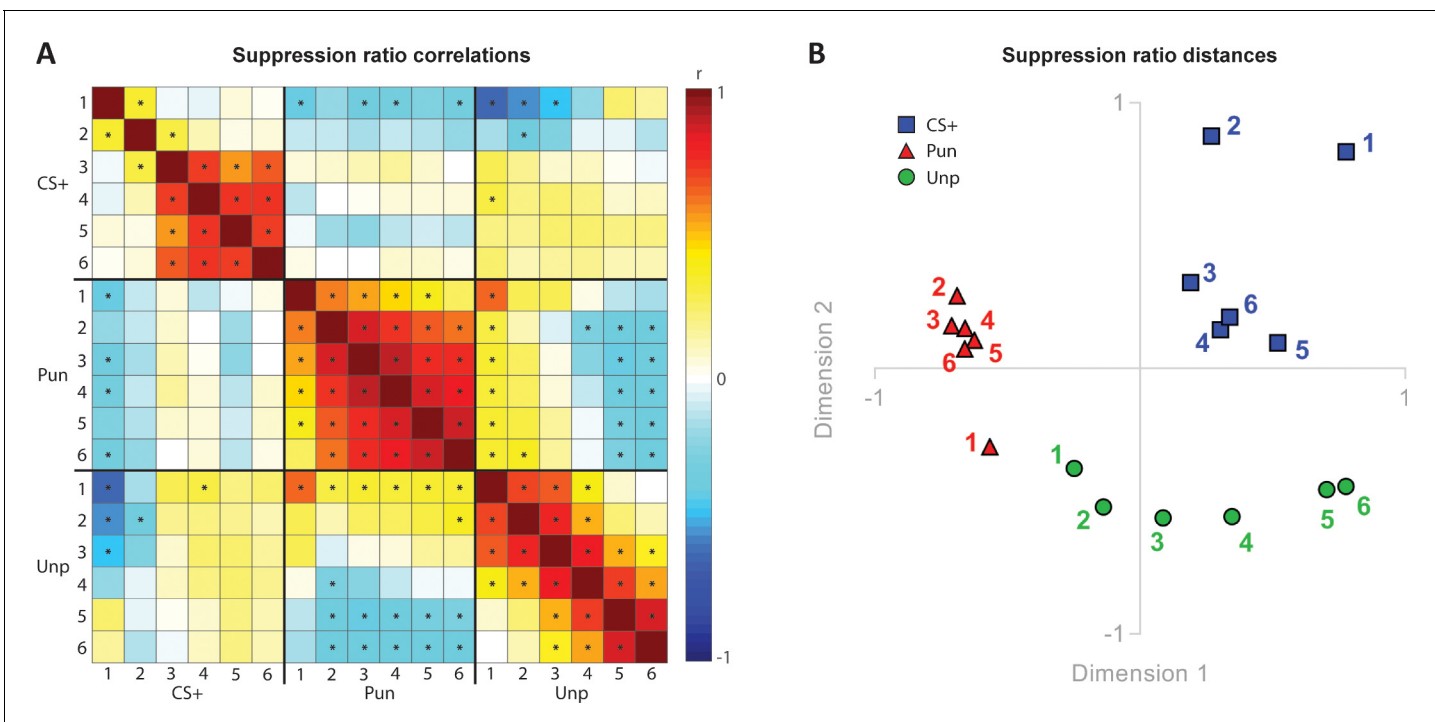

**Figure 4.** Relationships between punishment, fear and reward. (A) Correlation matrix for suppression ratios during CS+ presentations, punished lever, and unpunished lever across conditioned punishment sessions (1-6). (B) Multidimensional scaling showing suppression ratio distances for CS+, punished lever suppression, and unpunished lever suppression across sessions (1-6).

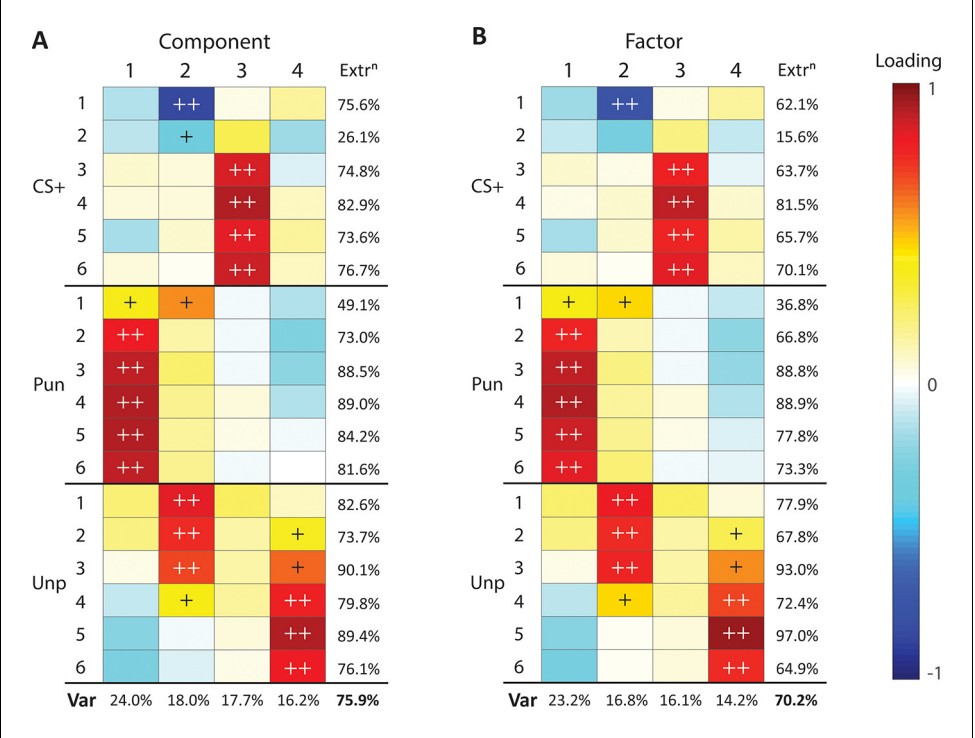

**Figure 5.** Principal component and factor analysis of suppression during conditioned punishment. (A) Loading heatmaps for principal component analysis of suppression ratios across conditioned punishment sessions (1-6). (B) Loading heatmaps for factor analysis of suppression ratios across conditioned punishment sessions (1-6). Bottom rows indicates proportion of total variance (Var) accounted for by components/factors. Last column indicates variance of each measure accounted for by components/factors (extraction). Loadings that account for majority (>50%) or substantial (>10%) variance are indicated with ++ and +, respectively.

The online version of this article includes the following figure supplement(s) for figure 5:

**Figure supplement 1.** Results of dimension reduction of suppression ratios.

**Figure supplement 2.** Relationships between rates of pre-punishment lever-pressing (T), ITI rates of lever-pressing (Pun LP, Unp LP) across conditioned punishment, and conditioned suppression.

later in training: only CS+ suppression loaded positively on this component. The fourth component captured reward: the remainder of the variance in unpunished responding loaded positively on this component. So, punishment and fear do not load positively on the same component. In fact, any relationship between them was largely negative, indicative of a competitive rather than complementary relationship between them.

PCA is a dimension reduction procedure. It is less explicitly a means to identify underlying latent variables in datasets. Therefore, we performed Factor Analysis to identify latent variables in the association between punishment, reward, and fear. The results from this analysis were similar to PCA (*Figure 5B*, *Figure 5—figure supplement 1*). Based on factor loadings, variation in aversive learning can be accounted for by an influence of punishment (Factor 1), contextual fear (Factor 2), CS+ fear (Factor 3), and reward (Factor 4). Taken together, these findings suggest punishment, reward, and fear are largely orthogonal each other.

We also assessed the relationship between pre-punishment lever-pressing and behaviour in conditioned punishment. Training lever-pressing was correlated with unpunished (average $r = 0.561$, p<0.0001) but not punished lever-press rates during ITIs (average $r = 0.198$, p=0.18) or conditioned suppression (average $r = 0.173$, p=0.24). This relationship was further supported by PCA and multidimensional scaling (*Figure 5—figure supplement 2*). This implies responding during training predicts later unpunished responding but not punishment or conditioned suppression. Lever-press suppression ratios, which remove variability attributable to pre-punishment differences in reward-seeking, were not correlated with training lever-press rate (punished lever suppression: $r = -0.06$,

p=0.69; unpunished lever suppression: $r = -0.04$, p=0.79), again indicating that punishment-driven changes in lever-pressing are unrelated to initial rates of lever pressing.

## Cluster analysis reveals punishment sensitive vs. insensitive phenotypes

To further understand individual differences in aversive learning we used cluster analysis. This allowed us to identify clusters of subjects whose performances across training were more similar to each other and different to other clusters of subjects. Silhouette values revealed positive silhouette values for 2–4 k-mean clusters, which were each marginally higher compared to solutions using more clusters.

We examined punishment, reward, and fear behaviours for each of the cluster solutions. The groups produced by the 2-cluster solution did not differ in sex ($\chi^2$(1)=0.782, p=0.376; *Figure 6—figure supplement 1*). The two clusters did not differ in pre-punishment lever-pressing (all $F_{(1,46)} \leq$. 659, $p \geq$. 421; *Figure 6—figure supplement 1*), showing that they did not differ in reward learning prior to aversive learning. They were, however, distinguishable by their punishment avoidance, regardless of whether this was measured via punished lever suppression or preference ratio (*Figure 6A*, *Figure 6—figure supplement 1*). Specifically, there was a significant overall difference in punished lever suppression ($F_{(1,46)} = 105.96$, p<0.001, $\eta_p^2$ 0.697) (*Figure 6A*) and preference ratio ($F_{(1,46)} = 49.13$, p<0.001, $\eta_p^2$ 0.517) (*Figure 6—figure supplement 1*) between clusters across sessions. However, there was no main effect of cluster on either unpunished lever suppression ($F_{(1,46)} = .215$, p = 0.645) (*Figure 6A*) or conditioned suppression ($F_{(1,46)} = 1.008$, p=0.321, $\eta_p^2$ 0.021) (*Figure 6B*). Thus, we refer to these clusters as punishment-sensitive (filled symbols, n = 15 [7 female]) and punishment-insensitive (empty symbols, n = 33 [11 female]). Further analyses showed that the punishment-sensitive group significantly suppressed punished ($F_{(1,46)} = 333.638$, p<0.001, $\eta_p^2$ 0.878) but not unpunished ($F_{(1,46)} = 2.601$, p=0.114, $\eta_p^2$ 0.053) responding relative to pre-punishment. In contrast, the punishment-insensitive group modestly suppressed both punished ($F_{(1,46)} = 75.318$, p<0.001, $\eta_p^2$ 0.621) and unpunished ($F_{(1,46)} = 10.395$, p=0.002, $\eta_p^2$ 0.184) responding relative to pre-punishment. This shows that the punishment-insensitive group were not simply showing attenuated punishment but were instead showing a distinct suppression phenotype.

Due to this differential ITI suppression across groups, shock intensities for the punishment-insensitive cluster had been increased more than for punishment-sensitive cluster (linear x cluster interaction: $F_{(1,46)} = 6.062$, p=0.018; *Figure 6—figure supplement 1*), although shock intensity did not differ overall across sessions ($F_{(1,45)} = 2.196$, p=0.145). Importantly, shock intensity was not a

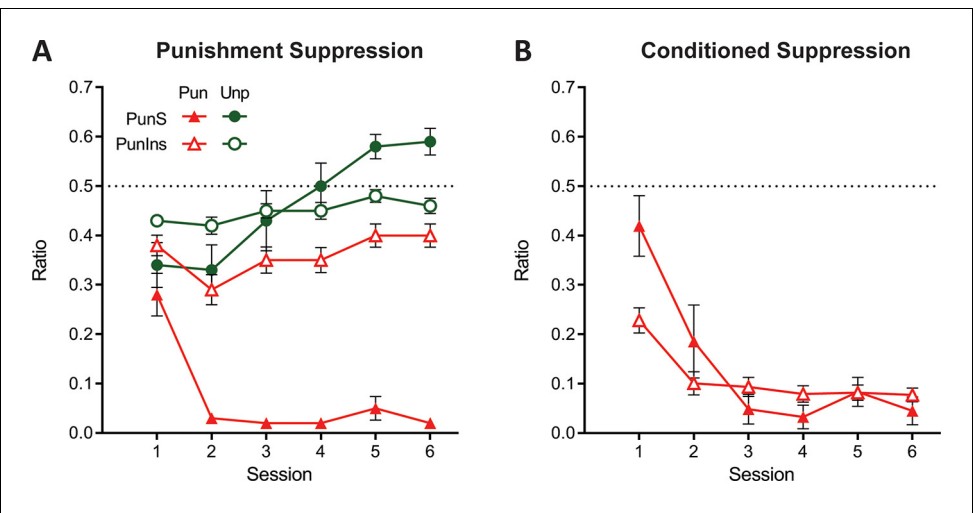

**Figure 6.** Behaviour of groups from 2-cluster solution. (**A**) Mean ± SEM punished and unpunished lever suppression for punishment-sensitive (PunS; filled) and punishment-insensitive (PunIns; empty) groups from 2-cluster solution. (**B**) Mean ± SEM conditioned suppression ratios for groups from 2-cluster solution.
The online version of this article includes the following figure supplement(s) for figure 6:

**Figure supplement 1.** Other results of 2-cluster solution.

significant covariate for final punishment (F(1,45) = 1.042, p=0.313) or conditioned suppression (F (1,45) = .389, p = 0.536), showing this was not a driving factor for cluster differences.

When a 3-cluster solution was derived, a significant effect of sex was found ($\chi^2$ (2)=7.416, p=0.025; *Figure 7—figure supplement 1*). The two larger clusters were most distinguishable by their suppression on the punished lever, that is punishment-sensitive (filled symbols; n = 17 [6 female]; *Figure 7A*) versus punishment-insensitive (empty symbols; n = 27 [8 female]; *Figure 7B*) and their behaviour was largely similar to the groups in the 2-cluster solution. The last cluster (half-filled symbols; n = 4 [4 female]; *Figure 7C*) was a small cohort that exhibited initial indiscriminate suppression on both the punished and unpunished levers as well as a counterintuitive increase in pressing during the CS+ during initial sessions. However, in later sessions, this cluster exhibited the greatest conditioned and punished lever suppression. Given these extreme responses, we will refer to this cluster as the hyper-sensitive group.

Once again, the clusters did not differ in lever-press rates across training (all F(2,45) $\leq$ 1.886, p $\geq$. 164; *Figure 7—figure supplement 1*). Compared to pre-punishment lever-pressing, all clusters showed significant punished lever suppression averaged across punishment (F(1,45) $\geq$ 45.374, p<0.001, $\eta_p^2$ 0.502) (*Figure 7*). However, there were differences between the clusters. Specifically, the punishment-insensitive cluster showed the least (F(1,45) $\geq$ 66.55, p<0.001, $\eta_p^2$ 0.596) and the hyper-sensitive cluster showed the most F(1,45) $\geq$ 4.386, p $\leq$. 0419, $\eta_p^2$ 0.089) punished lever suppression. The clusters also differed on unpunished lever suppression relative to pre-training (F(1,45) $\geq$ 15.54, p<0.001, $\eta_p^2$ 0.257) (*Figure 7*). The punishment-sensitive cluster showed no (F(1,45) = 1.164, p=0.286, $\eta_p^2$ 0.025), the punishment-insensitive cluster showed moderate (F(1,45) = 24.828, p<0.001, $\eta_p^2$ 0.356), whereas the hyper-sensitive cluster showed the most suppression (F(1,45) = 61.932, p<0.001, $\eta_p^2$ 0.579).

The three clusters showed different profiles of learning across days. Both the punishment-sensitive (F(1,45) = 13.84, p<0.001, $\eta_p^2$ 0.235) and hyper-sensitive clusters (F(1,45) = 20.471, p<0.001, $\eta_p^2$ 0.137) increasingly differentiated between punished and unpunished lever suppression across sessions whereas the punishment-insensitive cluster did not (F(1,45) = .168, p = 0.684). The punishment-sensitive cluster exhibited differential suppression for all sessions (F(1,45) $\geq$ 18.585, p < 0.001, $\eta_p^2$ 0.292), whereas the hyper-sensitive cluster only showed significantly different lever suppression from session four onwards (F(1,45) $\geq$ 15.088, p<0.001, $\eta_p^2$ 0.251). The punishment-sensitive cluster also initially suppressed (session 1–2: F(1,45) $\geq$ 5.934, p $\leq$. 019, $\eta_p^2$ 0.117) but subsequently elevated (session 3–6: F(1,45) $\geq$ 4.129, p $\leq$. 048) rates of unpunished responding relative to pre-punishment training. The hyper-sensitive cluster drastically suppressed unpunished responding early during aversive training (session 1–4: F(1,45) $\geq$ 19.060, p<0.001, $\eta_p^2$ 0.298), but this recovered and they eventually pressed at rates equivalent to pre-punishment training (session 5–6: F(1,45) $\leq$. 070, p $\geq$. 793).

All clusters showed greater CS+ fear across sessions (F(1,45) $\geq$ 6.08, p $\leq$. 018) (*Figure 7*). There were no significant differences between punishment-sensitive and punishment-insensitive clusters in their conditioned suppression (overall: F(1,45) = .517, p = 0.476, $\eta_p^2$ 0.011; linear: F(1,45) = .048, p = 0.828). However, the hyper-sensitive cluster had a significantly greater decrease in conditioned suppression across sessions than the other clusters (F(1,45) $\geq$ 41.255, p<0.001, $\eta_p^2$ 0.478).

The clusters differed in shock increments across training (linear x cluster interaction: F(2,45) = 6.850, p=0.003; *Figure 7—figure supplement 1*). However, shock intensity was not a significant covariate for final punishment (F(1,44) = 1.691, p=0.200) or conditioned suppression (F(1,44) = .834, p = 0.366), indicating that differences in shock intensity were not a driving factor for group differences.

## Discussion

Although punishment is highly conserved across species, it is far from robust across individuals. Here we studied individual differences in punishment sensitivity in rats, using a task permitting concurrent assessment of punishment, reward, and fear learning. We identified pronounced individual differences in punishment sensitivity. Using data-driven analytic approaches we show that these individual differences in punishment sensitivity cannot be predicted or explained by individual differences in fear or reward. Rather, across each analysis, punishment, fear and reward were remarkably

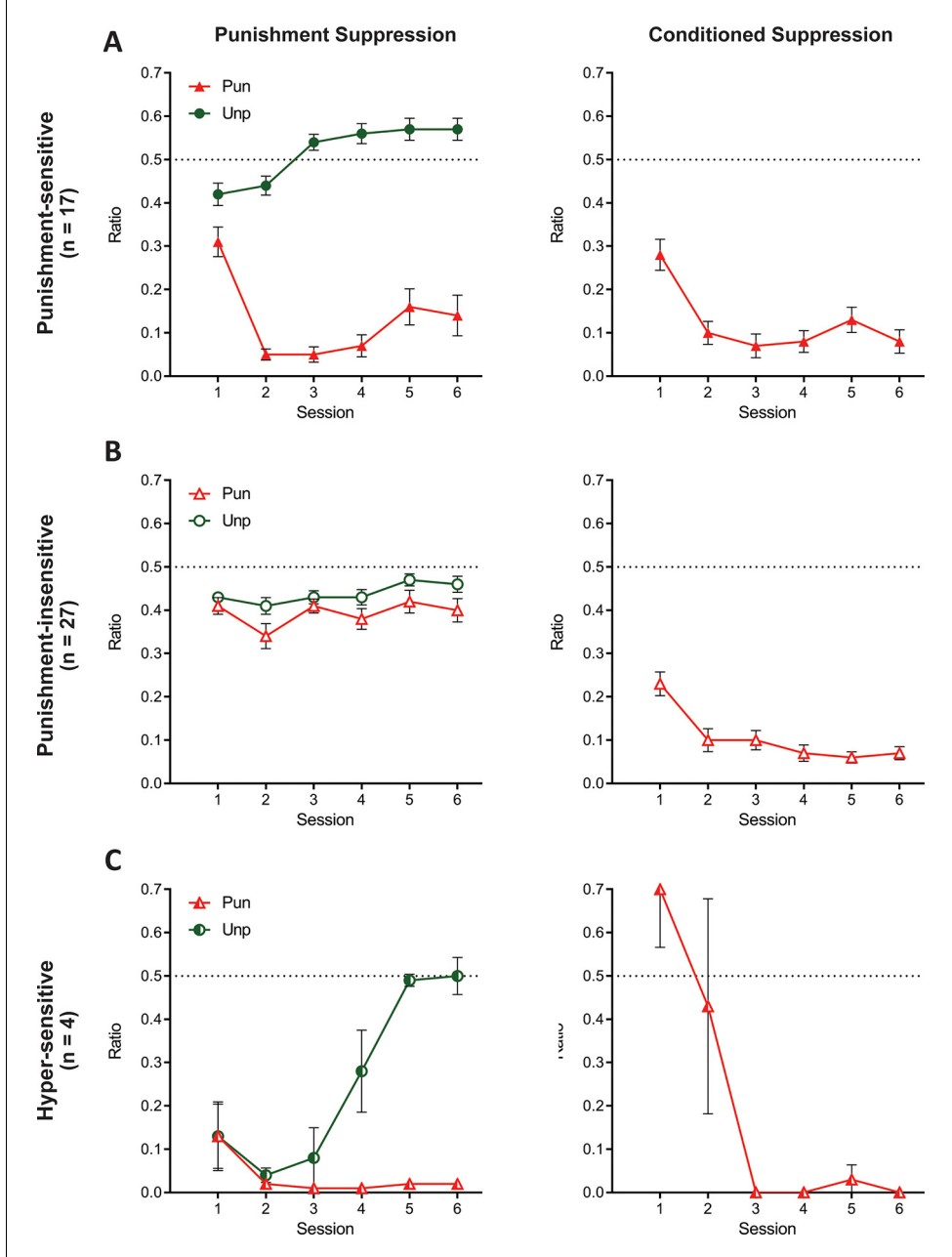

**Figure 7.** Behaviour of groups from 3-cluster solution. (A) Mean ± SEM punishment suppression and conditioned suppression for punishment-sensitive cluster. (B) Mean ± SEM punishment suppression and conditioned suppression for punishment-insensitive cluster. (C) Mean ± SEM punishment suppression and conditioned suppression for hyper-sensitive cluster.

The online version of this article includes the following figure supplement(s) for figure 7:

**Figure supplement 1.** Other results of 3-cluster solution.

independent. There was no evidence here to support the possibility that punishment insensitivity is due to reduced aversion sensitivity or reward dominance.

Instead, punishment insensitivity was a failure of instrumental learning. It could have multiple origins but is most likely due to a failure to encode the instrumental response-punisher association relative to the other associations in the task. Our task involved multiple instrumental (response-outcome) and Pavlovian (stimulus-outcome) contingencies; punishment-sensitive subjects parsed

these contingencies to show Pavlovian and instrumental behavioral control whereas insensitive subjects were impaired in partitioning these different contingencies. The strongest evidence for this possibility comes from the cluster analyses. Punishment-insensitive subjects identified by cluster analysis exhibited modest suppression of both punished and unpunished responding. This profile of generalised behavioural suppression is incompatible with the reward dominance account, and is similar to the behaviour shown by subjects receiving response-independent aversive events (*Hunt and Brady, 1951*; *Jean-Richard-Dit-Bressel et al., 2018*). That is, punishment-insensitive animals behaved as though they were not causing the aversive events they were experiencing and instead expressed weak but generalised Pavlovian fear. Alternatively, the subjects may have encoded this instrumental association but been unable to inhibit their punished behaviour in accordance with this knowledge. However, any such failure of inhibition must have been specific to the punished response and not a failure of behavioural inhibition more generally (*Gray, 1982*; *Gray and McNaughton, 2000*) because punishment-insensitive subjects showed intact behavioural inhibition during Pavlovian fear.

It is worth noting that punishment-insensitive rats were a relatively large proportion of the sample. Given the evidence that impaired punishment contingency detection underpinned punishment insensitivity, it is likely the relatively lean punishment contingency applied here (VI60 sec CS+) was a key factor in determining the number of insensitive subjects. Future research examining the effect of this contingency on punishment sensitivity would be useful. Interestingly, insensitivity to punishment in drug seeking has been observed using tighter response-punisher contingencies (*Marchant et al., 2018*). An intriguing possibility is that drugs of abuse may promote punishment resistance by impairing punishment contingency detection. This is consistent with demonstrations that insensitivity to punishment might be reduced at high shock intensities (*Golden et al., 2017*) or after extended punishment training (*Cooper et al., 2007*). Further work is needed to assess this.

From a theoretical perspective, perhaps the most surprising finding here was the absence of any notable co-variance between instrumental and Pavlovian aversive learning, even when more powerful methods capable of detecting such underlying relationships were applied (e.g. PCA and FA). Historically, theories of associative learning and motivation have assumed that instrumental and Pavlovian determinants of behaviour share a common basis and that reinforcers have common motivational value that underpins these different forms of learning (*Mackintosh, 1983*; *Rescorla and Solomon, 1967*). These theories have derived strong support from the inter-changeability of outcomes as reinforcers for Pavlovian and instrumental learning. They remain a dominant approach to understanding aversive learning (*Cain and LeDoux, 2008*). However, it is now well understood that distinct processes govern instrumental versus Pavlovian reward value (*Balleine and Dickinson, 1998*; *Dickinson and Balleine, 2002*). Our findings extend this dissociation to aversive learning (see also *Giuliano et al., 2018*; *Pelloux et al., 2007*). We show that if there is any trans-contingency encoding of outcome value, then this contributes little to how animals learn punishment and fear. These two forms of aversive learning were independent of each other suggesting that the motivational underpinnings of Pavlovian fear and punishment are distinct.

Our findings have important implications for use of punishment sensitivity in assessing motivation. Punishment tasks are widely used to model the adverse consequences of drug seeking and measure motivation to engage in drug-seeking in the face of adverse consequences (*Augier et al., 2018*; *Kasanetz et al., 2013*; *Pascoli et al., 2015*; *Vanderschuren and Everitt, 2004*; *Vanderschuren et al., 2017*). The persistence of drug-seeking in the face of punishment (i.e. insensitivity to punishment) is invoked as an objective behavioural marker of addiction (*Deroche-Gamonet et al., 2004*; *Vanderschuren and Everitt, 2004*). This insensitivity is typically attributed to drug-induced plasticity promoting reward dominance or impulsivity. However, a key finding here is that insensitivity is a characteristic of punishment itself. Punishment insensitivity can emerge from a specific deficit in instrumental aversive learning and can be observed in studies using non-drug rewards. This suggests that punishment insensitivity can pre-exist any drug-induced plasticity promoting reward dominance or impulsivity and this pre-existing difference may provide one basis for persistence of reward seeking in the face of punishment.

Other clinical populations are characterised by differences in sensitivity to punishment. Increased sensitivity to punishment is characteristic of depressive disorders; these individuals show catastrophic, globalised reactions to punishment (*Elliott et al., 1996*; *Eshel and Roiser, 2010*). Cluster analyses here identified a hyper-sensitive phenotype that initially displayed pronounced and

indiscriminate suppression of behaviour, commensurate with pronounced Pavlovian fear, before showing exaggerated punishment and appropriate discrimination between punished and unpunished behaviour. The transition from fear to punishment among hypersensitive animals was rapid and occurred within two sessions. Moreover, although there were no sex differences among the sensitive and insensitive phenotypes, the hypersensitive cluster was comprised exclusively of females. The relatively small number of animals in this hypersensitive cluster preclude further analyses, but the data-driven/bottom-up approach used to identify this cluster of hypersensitive subjects could prove useful for further research.

In summary, we examined punishment-, fear- and reward-related learning and behaviour in a task that permits assessment each of these processes concurrently in the same animals. We observed pronounced variations in punishment learning. We also identified clinically relevant phenotypes of insensitivity and hyper-sensitivity to punishment. In each case, these individual differences in punishment sensitivity could be explained by failures to encode the instrumental response-punisher association, not by aversion insensitivity or reward dominance. Subjects insensitive to punishment were afraid of the punisher but were unable to change their behaviour to avoid it.

# Materials and methods

## Subjects

Subjects were experimentally naive adult male and female Long-Evans rats (N = 48, 18 females) supplied by the University of New South Wales (Sydney, NSW, Australia). This was a single group experiment, so N = n = 48. This group size was chosen based on past research (*Marchant et al., 2018*) suggesting that it would be sufficient to identify individual differences in punishment. Animals were housed in groups of four in ventilated racks in a temperature- and humidity-controlled room with a 12–12 hr light/dark cycle (lights on 07:00). Experiments were conducted during the light cycle. Animals were food restricted from 3 days prior to the experiment onwards (10–15 g food per day for males, 7–12 g for females) to maintain them at ~90% of free-feeding weight, with ad libitum access to water. All procedures were approved by the UNSW Animal Ethics Committee (AEC) and in accordance with the code set out by the National Health and Medical Research Council (NHMRC) for the treatment of animals in research.

## Apparatus

Behavioural procedures were conducted in standard operant chambers (24 [length] x 30 [width] x 21 cm [height] (Med Associates, St Albans, VT) housed within sound- and light-attenuating cabinets equipped with fans providing constant ventilation and low-level background noise. All events were controlled and recorded by MedPC IV software (Med Associates). CS+ and CS- were 10 s 3 kHz tone or 5 Hz flashing light, counterbalanced. Pellets (Bioserve, Biotechnologies) were delivered from a dispenser to a recessed magazine cup (5 × 5 cm); magazine entries were detected using infrared beams at the magazine opening. Retractable levers were located on each side of the magazine. Shocks (0.5 secs, 0.3–0.6mA) were delivered via the grid floor. A 3W house light was mounted at the top of the wall opposite to the magazine and was turned on throughout each session.

**Table 1.** Experimental design.

| Lever | End lever-press training | Conditioned punishment |
|---|---|---|
| Punished | Food (VI30s) | Food (VI30s)<br>CS+ → Shock (VI60s) |
| Unpunished | Food (VI30s) | Food (VI30s)<br>CS- (VI60s) |

CS+ and CS- were 10 s 3 kHz tone or 5 Hz flashing light, counterbalanced. CS+ co-terminated with shock (0.5 secs, 0.3–0.6mA).

## Behavioural procedures (*Table 1*)

### Magazine training

Rats received one session of magazine training, during which pellets were delivered on a variable 60 s interval (VI-60s) schedule until 20 pellets were delivered or 30 min had elapsed.

### Lever-press training

Following magazine training, rats were trained to press two levers equally on an escalating reinforcement schedule. The first two sessions (30 mins) presented a single lever (left or right, order counterbalanced) to each rat and each lever-press was rewarded with a pellet (FR1). The session terminated after 20 presses or after 30 mins. Animals ($n$ = 2) that did not acquire lever-pressing received extra magazine and FR1 training. This was followed by single-lever sessions (30 mins) that reinforced lever-pressing on VI-15s and VI-30s schedule (one session for each schedule on each lever). Rats were then given double-lever sessions (30 mins); both levers were extended and reinforced on a VI-15s (one session) and modified VI-30s schedule (two sessions). To counteract lever-preferences and equalise lever-pressing on both levers, double-lever VI-30s sessions dynamically adjusted the VI schedule as a ratio of relative lever-press rates, decreasing the reinforcement schedule on the preferred lever and increasing the reinforcement schedule on the non-preferred lever. The last lever-press training session (60 mins) presented both levers and pressing was reinforced on a standard VI-30s schedule for each lever.

### Punishment and fear conditioning

Following lever-press training, rats received 6 days of conditioned punishment training. Both levers were extended for 60mins and pressing was reinforced on a standard VI-30 schedule. In these sessions, the punished lever also yielded an aversive CS+ (VI-60s), while pressing the other unpunished lever yielded a neutral CS- (VI-60s). The CS+ co-terminated with a 0.5 s footshock and the CS- terminated by itself. For the first session, footshock intensity was set at. 3mA. Shock intensity was intermittently incremented by. 1mA between sessions (up to. 6mA) if suppression of ITI lever-pressing was not observed. If a lever-press was scheduled to yield both a pellet and CS at the same time, only the CS was delivered due to its leaner schedule.

## Data analysis

Suppression/preference ratios were calculated using rates of lever-pressing on punished and unpunished levers during the inter-trial interval (ITI; non-CS periods), CS+, and CS-. Punishment learning was defined as suppression of punished lever-pressing during the ITI. This was captured in two ways. The traditional assessment is to measure rates of punished responding relative to unpunished responding during the ITI using a 'preference ratio' ([Pun ITI rate/total ITI rate]; previously termed a 'punishment ratio'). Suppression of punished as well as unpunished responding were also assessed using 'lever suppression ratios' (session ITI rate/[training ITI rate + session ITI rate]), which capture rates of responding on each lever relative to rates on the final day of pre-punishment training. This allows separate assessment of punished vs. unpunished responding under punishment, clarifying lever preferences or lack thereof. Finally, CS suppression ratios were calculated to assess suppression of lever-pressing (both levers) during each CS relative to ITI (CS rate/[ITI rate + CS rate]). The CS+ ratio measures Pavlovian fear via conditioned suppression, while the CS- ratio acts as a control comparison.

All three ratios range from 0 to 1. A CS or lever suppression ratio below 0.5 indicates suppression of lever-pressing relative to baseline, a ratio above 0.5 indicates elevated lever-pressing, while a ratio of 0.5 indicates no change/suppression. In the case of the preference ratio, a ratio of 0.5 indicates no preference between levers during the ITI, while a ratio below 0.5 indicates avoidance of the punished lever relative to the unpunished lever.

Ratios were analysed using polynomial contrasts in PSY. Significant suppression/bias was determined via single mean tests against the null of 0.5. Differences in ratios between groups and levers, and how these developed over sessions, were assessed via between x within ANOVAs. Lever (punished vs. unpunished) identity and session (linear trend) were used as within-subject factors where applicable. Cluster was applied as a between-subjects factor where applicable.

All other analyses were conducted in SPSS 25. Relationships between punishment and conditioned suppression were assessed via correlations, principal component analysis (PCA) and factor analysis (FA). CS+ suppression, punished and unpunished lever suppression ratios across the 6 days of conditioned punishment were used as inputs to parsimoniously capture aversively-motivated changes in behaviour while controlling for non-aversion related differences in responding. PCA and FA results were varimax rotated to improve interpretability of components/factors. Relationships between lever-press rates and conditioned suppression across conditioned punishment and last day of lever-press training were also assessed using correlations and principal component analysis.

Similarity/dissimilarity of suppression ratios or lever-press rates across sessions were also conveyed using multidimensional scaling (SPSS PROXSCAL with the following parameters: simplex, interval transformation, squared Euclidean distance, z-scored). These parameters provided an excellent fit of the data (suppression ratios: normalized raw stress = 0.01831, S-Stress = 0.04746, DAF = 0.98169, Tucker's coefficient for congruence = 0.99080; lever-press rates/conditioned suppression: normalized raw stress = 0.02288, S-Stress = 0.05471, DAF = 0.97712, Tucker's coefficient for congruence = 0.98849).

K-means clustering was used to assess distinct suppression phenotypes. Silhouette values were obtained for 2–7 clusters, revealing marginally higher values for 2–4 cluster solutions. Distribution of sex across clusters was assessed via chi square. To determine the possible contribution of different shock intensities on suppression, a between (cluster) x within (session) ANOVA was conducted on shock intensities. To assess the role of shock intensity on suppression, a univariate ANOVA was conducted for punished lever and conditioned suppression on last day of conditioned punishment using shock intensity as a covariate and cluster as a fixed factor.

## Acknowledgements

This work was supported by grants from the Australian Research Council (DP190100482; DP170100075).

## Additional information

### Funding

| Funder | Grant reference number | Author |
| --- | --- | --- |
| Australian Research Council | DP190100482 | Gavan P McNally |
| Australian Research Council | DP170100075 | Gavan P McNally |

The funders had no role in study design, data collection and interpretation, or the decision to submit the work for publication.

### Author contributions

Philip Jean-Richard-dit-Bressel, Conceptualization, Resources, Data curation, Software, Formal analysis, Investigation, Visualization, Methodology, Project administration; Cassandra Ma, Investigation; Laura A Bradfield, Conceptualization, Supervision, Investigation, Methodology, Project administration; Simon Killcross, Conceptualization, Supervision, Funding acquisition, Project administration; Gavan P McNally, Conceptualization, Resources, Data curation, Supervision, Funding acquisition, Project administration

### Author ORCIDs

Philip Jean-Richard-dit-Bressel  https://orcid.org/0000-0002-0898-8987
Laura A Bradfield  http://orcid.org/0000-0003-3921-0745
Gavan P McNally  https://orcid.org/0000-0001-9061-6463

## Ethics

Animal experimentation: All procedures were approved by the UNSW Animal Ethics Committee (AEC) (ACEC16/160B) and in accordance with the code set out by the National Health and Medical Research Council (NHMRC) for the treatment of animals in research.

## Decision letter and Author response

Decision letter https://doi.org/10.7554/eLife.52765.sa1
Author response https://doi.org/10.7554/eLife.52765.sa2

## Additional files

### Supplementary files

• Source data 1. Raw data.

• Transparent reporting form

### Data availability

All data generated or analysed during this study are included in the manuscript in Figure 1. Source data files, for all figures, have been provided.

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
