## [Decision Letter]

**Acceptance summary:**

In this paper Jean-Richard-dit-Bressel and colleagues developed a task to dissociate individual variability in sensitivity to reward and punishment. They employ a variety of creative analyses of behavior using PCA and multidimensional scaling, cluster analyses, etc. to show that the individual variance in sensitivity to punishment (i.e., suppression of responding due to the production of an aversive outcome) is unrelated to variance in sensitivity to reward or perception of the aversive US. The reviewers agreed that the core idea behind the manuscript - to examine the interplay between reward, punishment, and fear - is an extremely important and hugely understudied question, of relevance to adaptive behavior generally and many psychiatric disorders in particular. The design, execution, and analysis were all excellent, convincingly showing diversity in individual responding. This paper is important to understanding both appetitive and aversive processes, their interaction, neural mechanisms, and especially potential contribution to mental illness.

**Decision letter after peer review:**

Thank you for submitting your article "Punishment insensitivity emerges from impaired contingency detection, not aversion insensitivity or reward dominance" for consideration by *eLife*. Your article has been reviewed by three peer reviewers, including Geoffrey Schoenbaum as the Reviewing Editor and Reviewer #1, and the evaluation has been overseen by a Reviewing Editor and Kate Wassum as the Senior Editor. The following individuals involved in review of your submission have agreed to reveal their identity: Michael A McDannald (Reviewer #2); Yavin Shaham (Reviewer #3).

The reviewers have discussed the reviews with one another and the Reviewing Editor has drafted this decision to help you prepare a revised submission.

Summary:

The authors developed a task to dissociate individual variability in sensitivity to reward and punishment. Hungry rats were trained to press a pair of levers for food reward for a number of days, culminating in a VI30 schedule. Subsequently a CS->shock punishment was instituted on one lever on a VI60 schedule. With training, the rats learned to press for reward, biased their behavior away from the punished lever, and also showed fear behaviors to the CS+ prior to shock. The authors employ a variety of creative analyses of behavior using PCA and multidimensional scaling, cluster analyses, etc to show that the individual variance in sensitivity to punishment (i.e., suppression of responding due to the production of an aversive outcome) is unrelated to variance in sensitivity to reward or perception of the aversive US.

The reviews were uniformly positive. The reviewers agreed that the core idea behind the manuscript – to examine the interplay between reward, punishment, and fear – is an extremely important and hugely understudied question, of relevance to adaptive behavior generally and many psychiatric disorders in particular. The design and execution was excellent as well, especially the creative analyses, which was thorough and well-captured the diversity in individual responding. This paper will be of interest to those interested in appetitive and aversive processes, their interaction, neural mechanisms, or contribution to mental illness.

Essential revisions:

Although each reviewer raises concerns, on discussion none were deemed to be essential to address in any particular way. Thus, as long as the authors make a good faith effort to respond to the various points, we believe the manuscript will be publishable.

Reviewer #1:

In this study, the authors develop a task in which to dissociate individual variability in sensitivity to reward and punishment. Hungry rats were trained to press a pair of levers for food reward for a number of days, culminating in a VI30 schedule. Subsequently a CS->shock punishment was instituted on one lever on a VI60 schedule. With training, the rats learned to press for reward, biased their behavior away from the punished lever, and also showed fear behaviors to the CS+ prior to shock. The authors employ a variety of creative analyses of behavior using PCA and multidimensional scaling, cluster analyses, etc to show that the individual variance in behavioral measures of reward, punishment and fear were largely unrelated. They conclude that sensitivity to punishment (ie suppression of responding due to the production of an aversive outcome) is unrelated to variance in sensitivity to reward or perception of the aversive US.

Overall I thought the experiment and associated analyses were really excellent and exceptionally creative, and I think the results nicely support the authors' main point in the abstract that: "punishment insensitivity is (can be?) a unique phenotype, unrelated to differences in reward-seeking and Pavlovian fear".

As suggested by the added parentheses above, my main criticism is with the generalizability of the results. Specifically do the authors think their results mean that this is the only cause of behavior that ignores bad outcomes, or do they see their result as showing the clear and dissociable operation of one cause, without necessarily excluding others as potential causes? If the former, then I think they have a lot more work to do to rule out other possible causes, particularly when considering things such as addiction and depression.

But my guess is that they mean the latter, and currently they do use language in a number of places that suggest they are more circumspect. I think they just need to be more clear about this. In particular, it seems to me that they have a task that very nicely isolates a particular influence. And they show that influence can be dissociated from other potential causes of insensitivity. But I think they do not rule out a role for insensitivity in the other dimensions in the general ability to control behavior. Perhaps some of the features of their task minimize these influences? For instance, the rats are food deprived, well-trained on the food-seeking task, and on a particular schedule. They are also trained first for the food. These are all interesting variables that may make the reward-seeking more robust or dissociated from the subsequent effect of punishment. Likewise, the fear measure is taken during a CS that comes after the decision to respond. These are interesting decisions in arranging their test behavior, and in some regards, they might even make sense for modeling something like addiction where the reward is well-learned. But they might affect the results, I think. If shock was trained first or if the CS+ were presented prior to the choice, as in a transfer task, might the other mechanisms are be important in the control of behavior by aversive outcomes?

Of course, I imagine the authors think what I am suggesting is obvious. But the discussion and paper to do not come across like this. It seems to me that some consideration of these dimensions and pointing out the possibility of unique subgroups that could be revealed by other training protocols might be important. Otherwise I think the paper could be misunderstood by those less versed in behavior than the authors.

Reviewer #2:

In this study, Jean-Richard-dit-Bressel and colleagues examined the relationship between conditioned fear, punishment and reward in rats. This was done using a conditioning procedure in which rats were presented with two levers, each producing reward, but with one producing an auditory CS+ and the other an auditory CS-. Punishment was measured by the rat's propensity to press the lever producing the CS- over the CS+. Conditioned suppression was measured by the change in rate of pressing during cue presentation. The results were clear and compelling. All rats acquired conditioned suppression to the CS+, albeit with some individual variance. Punishment was also observed at the group level, but markedly more variability was observed. Subsequent analyses with multi-dimensional scaling and principal components demonstrated that punishment and conditioned suppression were largely unrelated, or even negatively correlated. Even more, individuals with could be classified and punishment-sensitive or insensitive. All in all, the results reveal independent behavioral mechanisms for reward, punishment and suppression.

All aspects of this study were excellent. The core idea behind the manuscript – to examine the interplay between reward, punishment is suppression is novel and simple. Yet testing these relationships is of obvious importance. Each process is essential to adaptive behavior and each is implicated in array of psychiatric disorders. The execution and analysis of the resulting data was thorough and well-captured the diversity in individual responding. Particularly elegant was the common scale on which punishment and suppression were measured. I believe the results are important and will garner considerable interest.

There are two areas in which I feel the manuscript could be strengthened and I have one comment concerning the language used to describe the procedure. These are provided below.

Measuring conditioned suppression

I had difficulty finding which lever was used to measure conditioned suppression. Were baseline and cue rates taken from both trial types? Or was only the CS- lever used because pressing was biased towards this lever? Specifically, I could not find a description of the lever used for the main result:

"There was also robust evidence for Pavlovian fear (Figure 2B). Conditioned suppression elicited by presentations of the CS+ also increased across training (F (1,47) = 35.1; p <.001, ηp2 = 0.427), with significant suppression being observed for each session (all F (1,47) > 54.7; p <.001, η_p_^2^ = 0.537)."

I also could not determine which lever was being used for suppression in the main figure (Figure 2).

If the authors could specify exactly which lever(s) was used for calculating suppression ratio it would be helpful.

Examining reward

I was very impressed with the author's treatment of conditioned suppression and punishment, both of which were well captured by use of a ratio. However, reward responding seemed less well captured by this measure. At times, I could not determine what measure was being used to assess reward responding. I realize this is difficult while the conditioning procedure is ongoing, as biases in lever pressing should be observed during cue and ITI periods.

However, a pure measure of lever-pressing is available in the lever-press training prior to the beginning of fear discrimination. I am curious if lever press rates observed during this time predict performance in punishment and suppression. For example, rats showing high press rates prior to discrimination may be punishment-insensitive rats OR these rats may show less conditioned suppression. These relationships could be initially examined with simple tests like Pearson's correlation coefficient. This would provide a clearer and more direct examination of the relationship between reward and conditioned punishment & conditioned suppression. If relationships are found, multi-dimensional scaling and principal components could be performed with this factor.

Punishment vs. Conditioned Punishment

The Abstract and Introduction describe the impetus of the study to disentangle reward, punishment and Pavlovian fear (suppression). For the most part, this is reasonable and sets up the reader for the study performed. As the authors are aware (indeed, Dr. Killcross is an author) this procedure was initially designed to dissociate conditioned suppression from conditioned punishment (Killcross et al., 1997). Punishment and conditioned punishment are likely to require independent + overlapping neural and behavioral mechanisms. For this reason, I think it would be prudent to state in the Abstract that conditioned punishment is measured. This should also be stated at the end of the Introduction – when the behavioral procedure is discussed. Indeed, when I first started reading the manuscript, I assumed direct punishment was going to be assessed. Ultimately, I think the use of conditioned punishment – as the authors performed – was more appropriate. Making this clear at the outset of the manuscript will better prepare the reader for the experiment that was performed.

Reviewer #3:

This is an excellent paper in which the authors used a creative two-lever operant procedure to study individual differences in punishment responding and the relationship between responding to punishment of food reward, conditioned fear responding to the punishment cue, and responding for food reward. The main finding is that punishment responding is unrelated to either conditioned fear or food reward responding. The main important general conclusion is that punishment insensitivity is not due to either reduced aversion sensitivity or higher reward value. The authors proposed that punishment insensitivity reflects a failure to learn instrumental control over punishment.

Overall, the behavioral procedure is elegant, the behavioral effects appear robust and reproducible, the experimental methodology is sound, and the statistical analyses are appropriate to the experimental design and research questions. The paper is also very well written and includes appropriate historical citations. I enclose below several comments.

1) The surprising finding in the study was the large number of punishment insensitive rats in the authors' procedure (22/30 males, 11/18 females). Typically, in punishment studies, with increased shock intensity all subjects eventually learn the punishment task. The authors should discuss this issue in the revision. In future studies, the authors should consider manipulating shock intensity parametrically to generate a more sensitive measure of punishment (the equivalent of ED50 in pharmacological dose-response curve) to characterize individual differences in punishment.

2) Subsection “Data analysis”: Change "inter-trial period" to "inter-trial-interval" to fit the abbreviation ITI.

3) Results section: Please add the final shock intensity value for the punishment sensitive and insensitive groups. I presume it was higher for the punishment insensitive group, but this was not described.

---

## [Author Response]

Reviewer #1:In this study, the authors develop a task in which to dissociate individual variability in sensitivity to reward and punishment. Hungry rats were trained to press a pair of levers for food reward for a number of days, culminating in a VI30 schedule. Subsequently a CS->shock punishment was instituted on one lever on a VI60 schedule. With training, the rats learned to press for reward, biased their behavior away from the punished lever, and also showed fear behaviors to the CS+ prior to shock. The authors employ a variety of creative analyses of behavior using PCA and multidimensional scaling, cluster analyses, etc to show that the individual variance in behavioral measures of reward, punishment and fear were largely unrelated. They conclude that sensitivity to punishment (ie suppression of responding due to the production of an aversive outcome) is unrelated to variance in sensitivity to reward or perception of the aversive US.Overall I thought the experiment and associated analyses were really excellent and exceptionally creative, and I think the results nicely support the authors' main point in the abstract that: "punishment insensitivity is (can be?) a unique phenotype, unrelated to differences in reward-seeking and Pavlovian fear".As suggested by the added parentheses above, my main criticism is with the generalizability of the results. Specifically do the authors think their results mean that this is the only cause of behavior that ignores bad outcomes, or do they see their result as showing the clear and dissociable operation of one cause, without necessarily excluding others as potential causes? If the former, then I think they have a lot more work to do to rule out other possible causes, particularly when considering things such as addiction and depression.But my guess is that they mean the latter, and currently they do use language in a number of places that suggest they are more circumspect. I think they just need to be more clear about this. In particular, it seems to me that they have a task that very nicely isolates a particular influence. And they show that influence can be dissociated from other potential causes of insensitivity. But I think they do not rule out a role for insensitivity in the other dimensions in the general ability to control behavior. Perhaps some of the features of their task minimize these influences? For instance, the rats are food deprived, well-trained on the food-seeking task, and on a particular schedule. They are also trained first for the food. These are all interesting variables that may make the reward-seeking more robust or dissociated from the subsequent effect of punishment. Likewise, the fear measure is taken during a CS that comes after the decision to respond. These are interesting decisions in arranging their test behavior, and in some regards, they might even make sense for modeling something like addiction where the reward is well-learned. But they might affect the results, I think. If shock was trained first or if the CS+ were presented prior to the choice, as in a transfer task, might the other mechanisms are be important in the control of behavior by aversive outcomes?Of course, I imagine the authors think what I am suggesting is obvious. But the discussion and paper to do not come across like this. It seems to me that some consideration of these dimensions and pointing out the possibility of unique subgroups that could be revealed by other training protocols might be important. Otherwise I think the paper could be misunderstood by those less versed in behavior than the authors.

We agree that protocol parameters are certainly a factor. A deeper examination of the effects that changing these parameters might have on learning and behaviour, including the subgroups they might reveal, is an interesting area for future research. This has been added to the Discussion section.

Reviewer #2:In this study, Jean-Richard-dit-Bressel and colleagues examined the relationship between conditioned fear, punishment and reward in rats. This was done using a conditioning procedure in which rats were presented with two levers, each producing reward, but with one producing an auditory CS+ and the other an auditory CS-. Punishment was measured by the rat's propensity to press the lever producing the CS- over the CS+. Conditioned suppression was measured by the change in rate of pressing during cue presentation. The results were clear and compelling. All rats acquired conditioned suppression to the CS+, albeit with some individual variance. Punishment was also observed at the group level, but markedly more variability was observed. Subsequent analyses with multi-dimensional scaling and principal components demonstrated that punishment and conditioned suppression were largely unrelated, or even negatively correlated. Even more, individuals with could be classified and punishment-sensitive or insensitive. All in all, the results reveal independent behavioral mechanisms for reward, punishment and suppression.All aspects of this study were excellent. The core idea behind the manuscript – to examine the interplay between reward, punishment is suppression is novel and simple. Yet testing these relationships is of obvious importance. Each process is essential to adaptive behavior and each is implicated in array of psychiatric disorders. The execution and analysis of the resulting data was thorough and well-captured the diversity in individual responding. Particularly elegant was the common scale on which punishment and suppression were measured. I believe the results are important and will garner considerable interest.There are two areas in which I feel the manuscript could be strengthened and I have one comment concerning the language used to describe the procedure. These are provided below.Measuring conditioned suppressionI had difficulty finding which lever was used to measure conditioned suppression. Were baseline and cue rates taken from both trial types? Or was only the CS- lever used because pressing was biased towards this lever? Specifically, I could not find a description of the lever used for the main result:"There was also robust evidence for Pavlovian fear (Figure 2B). Conditioned suppression elicited by presentations of the CS+ also increased across training (F (1,47) = 35.1; p <.001, ηp2 = 0.427), with significant suppression being observed for each session (all F (1,47) > 54.7; p <.001, η_p_^2^ = 0.537)."I also could not determine which lever was being used for suppression in the main figure (Figure 2).If the authors could specify exactly which lever(s) was used for calculating suppression ratio it would be helpful.

Response rate across both levers were used to determine conditioned suppression. Text in the Materials and methods section has been changed to make this clearer.

Examining rewardI was very impressed with the author's treatment of conditioned suppression and punishment, both of which were well captured by use of a ratio. However, reward responding seemed less well captured by this measure. At times, I could not determine what measure was being used to assess reward responding. I realize this is difficult while the conditioning procedure is ongoing, as biases in lever pressing should be observed during cue and ITI periods.However, a pure measure of lever-pressing is available in the lever-press training prior to the beginning of fear discrimination. I am curious if lever press rates observed during this time predict performance in punishment and suppression. For example, rats showing high press rates prior to discrimination may be punishment-insensitive rats OR these rats may show less conditioned suppression. These relationships could be initially examined with simple tests like Pearson's correlation coefficient. This would provide a clearer and more direct examination of the relationship between reward and conditioned punishment & conditioned suppression. If relationships are found, multi-dimensional scaling and principal components could be performed with this factor.

This is an important question. We had previously addressed this indirectly by assessing differences between clusters in lever-press rates at the end of training (they did not differ). However, we agree that this analysis was incomplete. To provide a more direct assessment of effects related to pre-punishment responding, we have added a paragraph to the Results section and Figure 5—figure supplement 2 analysing relationships between lever-pressing at end of training and behaviour under conditioned punishment. In summary, lever-press rates prior to punishment could not predict punishment or conditioned suppression but did predict unpunished responding. We have also updated the supplementary figures for cluster analyses (Figure 6—figure supplement 1, Figure 7—figure supplement 1) to show lever-press rates *across* lever-press training to give a more complete picture of potential differences between groups. Again, none were observed.

Punishment vs. Conditioned PunishmentThe Abstract and Introduction describe the impetus of the study to disentangle reward, punishment and Pavlovian fear (suppression). For the most part, this is reasonable and sets up the reader for the study performed. As the authors are aware (indeed, Dr. Killcross is an author) this procedure was initially designed to dissociate conditioned suppression from conditioned punishment (Killcross et al., 1997). Punishment and conditioned punishment are likely to require independent + overlapping neural and behavioral mechanisms. For this reason, I think it would be prudent to state in the Aabstract that conditioned punishment is measured. This should also be stated at the end of the Introduction – when the behavioral procedure is discussed. Indeed, when I first started reading the manuscript, I assumed direct punishment was going to be assessed. Ultimately, I think the use of conditioned punishment – as the authors performed – was more appropriate. Making this clear at the outset of the manuscript will better prepare the reader for the experiment that was performed.

We agree and have added conditioned punishment has been added to Abstract and end of Introduction.

Reviewer #3:This is an excellent paper in which the authors used a creative two-lever operant procedure to study individual differences in punishment responding and the relationship between responding to punishment of food reward, conditioned fear responding to the punishment cue, and responding for food reward. The main finding is that punishment responding is unrelated to either conditioned fear or food reward responding. The main important general conclusion is that punishment insensitivity is not due to either reduced aversion sensitivity or higher reward value. The authors proposed that punishment insensitivity reflects a failure to learn instrumental control over punishment.Overall, the behavioral procedure is elegant, the behavioral effects appear robust and reproducible, the experimental methodology is sound, and the statistical analyses are appropriate to the experimental design and research questions. The paper is also very well written and includes appropriate historical citations. I enclose below several comments.1) The surprising finding in the study was the large number of punishment insensitive rats in the authors' procedure (22/30 males, 11/18 females). Typically, in punishment studies, with increased shock intensity all subjects eventually learn the punishment task. The authors should discuss this issue in the revision. In future studies, the authors should consider manipulating shock intensity parametrically to generate a more sensitive measure of punishment (the equivalent of ED50 in pharmacological dose-response curve) to characterize individual differences in punishment.

We agree, the number of punishment-insensitive animals was indeed surprising. We believe the relatively weak response-punisher contingency is a reason so many subjects failed to acquire punishment avoidance, despite the relatively high shock intensity. Text has been added to the Discussion section to note this.

2) Subsection “Data analysis”: Change "inter-trial period" to "inter-trial-interval" to fit the abbreviation ITI.

"Inter-trial period" changed to "inter-trial-interval".

3) Results section: Please add the final shock intensity value for the punishment sensitive and insensitive groups. I presume it was higher for the punishment insensitive group, but this was not described.

Yes. Correct. Shock intensity per group has been added to cluster supplementary figure (Figure 6—figure supplement 1, Figure 7—figure supplement 1). This presents a slight confound, so we assessed the possible contribution of shock intensity as a covariate in punishment and conditioned suppression; shock intensity did not significantly co-vary with punishment or conditioned suppression.